# Implications and Emerging Therapeutic Avenues of Inflammatory Response in HPV+ Head and Neck Squamous Cell Carcinoma

**DOI:** 10.3390/cancers14215406

**Published:** 2022-11-02

**Authors:** Lúcio Roberto Cançado Castellano, Sara Brito Silva Costa Cruz, Michael Hier, Paulo Rogério Ferreti Bonan, Moulay A. Alaoui-Jamali, Sabrina Daniela da Silva

**Affiliations:** 1Department of Otolaryngology and Head and Neck Surgery and Lady Davis Institutes for Medical Research of the Jewish General Hospital, McGill University, Montreal, QC H3T 1E2, Canada; 2Human Immunology Research and Education Group, Federal University of Paraíba, João Pessoa 58051-900, PB, Brazil; 3Graduate Program in Dentistry, Federal University of Paraíba, João Pessoa 58051-900, PB, Brazil

**Keywords:** papillomavirus infections, head and neck squamous cell carcinoma, tumor microenvironment, cellular immunity, immunotherapy

## Abstract

**Simple Summary:**

Cancer in the head and neck region (HNSCC) is exponentially increasing due to human papillomavirus (HPV) infections. This paper helps us to understand the complexity of the inflammatory networks and the mechanisms of immune evasion in HPV+ HNSCC to open up new avenues and drive the discovery of useful tools to be translated clinically in the screening and treatment of these cases, especially to overcome resistance and improve patients’ quality of life.

**Abstract:**

Head and neck squamous cell carcinomas (HNSCC) are a heterogeneous group of malignancies which have shown exponential incidence in the last two decades especially due to human papillomavirus (HPV) infection. The HPV family comprises more than 100 types of viruses with HPV16 and HPV18 being the most prevalent strains in HNSCC. Literature data reveal that the mutation profile as well as the response to chemotherapy and radiotherapy are distinct among HPV+ versus HPV-negative tumors. Furthermore, the presence of the virus induces activation of an immune response, in particular the recruitment of specific antiviral T lymphocytes to tumor sites. These T cells when activated produce soluble factors including cytokines and chemokines capable of modifying the local immune tumor microenvironment and impact on tumor response to the treatment. In this comprehensive review we investigated current knowledge on how the presence of an HPV can modify the inflammatory response systemically and within the tumor microenvironment’s immunological responses, thereby impacting on disease prognosis and survival. We highlighted the research gaps and emerging approaches necessary to discover novel immunotherapeutic targets for HPV-associated HNSCC.

## 1. Head and Neck Squamous Cell Carcinoma (HNSCC)

Head and neck squamous cell carcinoma (HNSCC) represents the sixth most frequent cancer worldwide [1]. HNSCC is etiologically associated to exposure to extrinsic carcinogens such as smoking and alcohol consumption [2,3]. Since the late 1990s, there has been an exponential rise in HNSCC incidence, especially oropharyngeal tumors (OPSCC) in countries with the highest median income, which is related to human papillomavirus (HPV) infection [4,5]. HPV has over 200 serotypes, with HPV16 and HPV18 strains being primarily responsible for HPV-related HNSCC [6]. HPV+ OPSCC is most common in healthier, younger, and non-smoking patients [5]. Significant progress has been achieved over the past few decades in the molecular profile and characterization of both HPV- and HPV+ HNSCC. These cooperative efforts established the impact of mutations in *TP53* (84%), *CDKN2A* (58%), *CCND1* (31%), and the overexpression of PI3K pathways (30%) in HPV-negative cases associated with tobacco, but fewer genomic alterations were observed in HPV+ HNSCC [4].

HPV+ OPSCC presents distinctive differences from HPV- tumors in the infiltrating immune cell population profile, underlining a unique biology of this malignancy [7,8,9]. The HPV infection is an early event and most of the HNSCC arises from deep lingual tonsils and palatine crypts. Reticulated crypt epithelium in the oropharynx is unique to this anatomical location in the head and neck, and may explain why HPV is estimated to be five times higher in the oropharynx when compared to the oral cavity, larynx, or hypopharynx [10]. During the course of the HNSCC’s development and progression, the tumor cells and the surrounding microenvironment (TME) are in constant communication and continuously evolve together [10,11]. Tumor cells can adapt several mechanisms to escape immune surveillance, favor tumor growth, proliferation, survival, and promote invasion [6,12,13]. In this scenario, the understanding of the inflammatory networks and the mechanisms of immune evasion in HPV+ HNSCC could help us to drive the discovery of useful tools to be translated clinically in the screening and treatment of these cases, especially to overcome resistance and improve patients’ quality of life.

## 2. Inflammatory Response in HNSCC

Inflammation in cancer has been described to initiate genetic instability in tumor cells [14]. During HPV infection, infiltrating immune cells interact with the virus to induce and/or activate epithelial cell differentiation [15]. The TME of HNSCC is composed of a heterogeneous cell population integrated in a complex extracellular matrix (ECM) [16]. The main cellular components of the TME are tumor-infiltrating lymphocytes (a.k.a.: TILs; or B and T lymphocytes), tumor-associated macrophages (TAMs), natural killer cells (NKs), tumor-associated neutrophils (TANs), dendritic cells (DCs), and cancer-associated fibroblasts (CAFs) [17,18] (Figure 1). In an initial stage, the tumor development can be enriched by cytotoxic innate lymphocytes (e.g., NKs) and adaptive immune cells (e.g., B and T lymphocytes); however, progressive cancer cells can regulate different signaling mechanisms that mimic immune tolerance in order to evade the tumoricidal attack and eventually lead to tumor metastasis [19]. 

The better prognosis of HPV+ HNSCC when compared to HPV- HNSCC has been associated with the higher number of TILs [20,21,22]. The TIL population can be classified into two major subsets: CD4+ and CD8+ T lymphocytes (Figure 2). Furthermore, the effector CD4+ T lymphocytes are subdivided into two groups with distinctive characteristics: regulatory T (Treg) and helper T (Th) cells [23]. In HNSCC, the CD8+ T lymphocyte infiltration has anti-tumoral activity and its presence is related to a favorable outcome [24]. In HPV+ HNSCC, the tumor antigen tolerance has been attributed to the presence of abundant levels of activated Treg lymphocytes within the TME [25,26,27]. However, effector T cells can polarize into exhausted T cells leading to cancer immune evasion [28]. T-cell exhaustion is a hyporesponsive state of T cells characterized by increased inhibitory receptors (such as: cytotoxic T lymphocyte antigen-4 (CTLA-4), programmed cell death protein 1 (PD-1), the T-cell immunoglobulin domain and mucin domain protein 3 (TIM-3), the lymphocyte activation gene 3 protein (LAG-3), the band T lymphocyte attenuator (BTLA), the T-cell immunoglobulin and immunoreceptor tyrosine-based inhibitory motif domain (TIGIT)), as well as the decreased effector cytokines (such as: interleukin-2 (IL-2), tumor necrosis factor-α (TNF-α), interferon-γ (IFN-γ), granzyme B (GzmB)), which impair the cytotoxicity leading to an inability to eliminate cancer cells [27,29,30,31,32,33,34]. Therefore, reversing the T-cell exhaustion status might represent a potential strategy to treat cancer. It is known that the secretion of immunesuppressive cytokines (such as the transforming growth factor-β (TGF-β) and interleukin-10 (IL-10)) is a major contributor to immune tolerance by regulatory T cells (Tregs) [35]. They are able to enhance tumor cell proliferation, survival, and metastasis by regulating anti-tumor immunity [36,37]. However, the therapeutic application of inhibitory cytokines remains a challenge to exploit their anti-tumor activity while keeping a low level of toxicity [38]. In the same way, multiple reports demonstrated the essential role of chemokines receptors (CCR4, CCR5, CXCR3, CXCR4, CCR6, and CCR7) in the regulation of Tregs for trafficking and homing for inflammatory sites in oral cell lines [39,40,41,42,43]. Chemokines are a superfamily of proteins that act as mediators not only affecting immune-cell infiltration into tumor sites, but also having a great impact on cancer progression by inducing ECM degradation via matrix metalloproteinase (MMP), and promoting neovascularization [44,45]. Chemokines are potentially dual-functional (homeostatic and inflammatory) during tumor development. The same chemokines can be either favorable or unfavorable prognostic indicators depending on the type and/or stage of the malignancies [45]. The contribution of chemokines to tumor progression depends on the balance between tumor-promoting and tumor-inhibiting factors [46]. Cytokines and chemokines may become formidable partners in synergistic therapeutic strategies combined with gene and/or cell therapy and monoclonal antibody-based therapies; however, further studies need to be carried out to confirm the safety and benefit over the current therapeutic strategy.

The major components in the TME are the tumor-associated macrophages (TAMs) and they are highly dynamic and heterogeneous and are tamed by tumor cells to promote tumor growth and progression [47]. TAMs can express cytokines that stimulate tumor cell proliferation and survival by regulating the transforming growth factor (TGF-β), the epithelial growth factor (EGF) as well as the EGF ligants and receptor (EGFR), the hepatocyte growth factor (HGF), the platelet-derived growth factor (PDGF), and the fibroblast growth factor (FGF) [48]. These cells also play a crucial role in the reorganization of the TME by promoting tumor cell motility via ECM degradation and the initiation of angiogenesis in hypoxic areas with a poor blood supply [49]. TAMs involve multiple phenotypes associated with a wide range of functions under distinctive pathological conditions. For instance, macrophages can be classified into groups depending on their activity and polarized status, as classically (M1) or alternatively activated (M2) [50]. The hypoxic tumor area contains chemokines and immunomodulatory proteins (such as CSF1, TGF-β, CCL2, FTL and FTH) which promote the polarization of TAMs into M2 macrophages [49]. In conventional cell-mediated immune responses, M1 macrophages have pro-inflammatory functions activated through IFN-γ, Th1 cytokines, and lipopolysaccharides in response to the presence of pathogens [51]. M1 TAM promotes the destruction of cancer cells and the inhibition of angiogenesis, concomitant with the activation of an inflammatory reaction [52]. Conversely, the anti-inflammatory and immunosuppressive cytokine-chemokine TME is responsible for dampening the macrophage activation by inducing its polarization towards the anti-tumor M2 profile [18,53]. M2 TAMs secrete pro-tumor factors (such as IL-4, IL-6, IL-10, IL-13, EGF, and VEGF), while the tumor suppressor M1 TAMs expresses anti-tumor factors (such as IL-12, the major histocompatibility complex (MHC) class II, and TNF-α) [53]. Furthermore, M2 macrophages may be responsible for inefficient tumor antigen presentation via MHC class I to cytotoxic cells such as NKs and CD8^+^ T lymphocytes [18,43]. The overexpression of polarized M2 TAMs and Tregs in HNSCC is associated with poor overall survival in HNSCC [11,17,25,26]. The unbalanced immune response at the tumor site interferes with the recruitment and activation of effector cells to support (or not) an efficient immune response to the tumor. The circulating Tregs’ and TAMs’ polarization status leading to increased expression of immune-suppressive cytokines is a fundamental mechanism by which the tumors can escape the immune surveillance, and they are also a promising target for further investigations as anti-cancer therapies. However, the conditions within the TME endow the immune cells with plasticity and versatility, and this dynamic interaction will define the specific functions and how the tumor will progress. 

## 3. HPV Infection and the Impact on the Immune System in HNSCC

HPV is a group containing more than 100 different types of viruses with pathogenic behavior to humans [6]. They are a circular, non-enveloped, double-stranded DNA (dsDNA) virus approximately 8kb in size that infects basal keratinocytes [54]. Their genome can be divided into an early region (E1, E2, E4, E5, E6, and E7, responsible for virus replication and generation of oncoproteins), a late region (L1 and L2 are the major and minor capsid or coat proteins responsible for structural components of the virus), the virus-like particle (VLP), and long control region (LCR, responsible for the virus transcription and the epithelial tropism) [55,56] (Figure 3). The stratified epithelium of the oral cavity is the target site for HPV to initiate infection. HPV entry is achieved via complex interactions of the viral capsid with cellular proteins leading to conformational changes within the capsid via proteases and chaperones, and interaction of the capsid proteins with different cell receptors. The key mechanisms in HPV entry and trafficking are still under scientific debate and studies on HPV infection mechanisms produce diverse and contradictory results. The discrepancies may partially be attributable to the different virus genotypes, cell lines, and methods of virus production that have been used for the experimental setups and to different observations from in vivo and in vitro models [57,58,59,60,61].

Among HPV types, HPV16 and HPV18 are considered as high risk (HR) HPV and they are detected in 90% of the HPV+ HNSCC patients [62,63,64]. A higher frequency of oral sex and casual sexual activity involving multiple partners are associated with the elevated risk of HPV-related cancer in HNSCC [65,66]. Following infection, the virus can remain in its episomal form, or become integrated into the host genome [67]. In the majority of cases, HPV infection is transient and it is solved spontaneously; however, in certain individuals, the viral clearance does not occur and the infection becomes persistent resulting in lesions that may eventually progress to cancer. The mechanisms of the clearance of HPV infection in some individuals remain unknown, but persistent infection with HR-HPV is necessary for tumor development [68]. Most often, the integration of HPV DNA hijacks the host cell genome to initiate viral DNA replication and amplification of their own genome [10,11]. This may result in genetic rearrangements, chromosomal inversions and translocations, gene deletions, the activation of proto-oncogenes and loss of heterozygosity, which generates genomic instability and increases the risk of neoplastic cell transformation through uncontrolled cell proliferation and resistance to death (Figure 3) [68].

The HPV persistent infection also requires a tolerant TME-supporting virus evasion and/or the suppression of the immunological responses [69]. The intratumor immune dynamic in HPV+ HNSCC is different from HPV-negative HNSCC [25,26,27,69,70,71], mainly due to the activity of viral proteins constantly stimulating the immune cell repertoire [72]. HPV oncoproteins interact with the host cells leading to (i) the integration of the virus into the host genome [73]; (ii) the induction of cell proliferation and differentiation; (iii) the host-cell immortalization [74]; (iv) the inhibition of apoptosis [75,76,77,78,79,80]; and (v) the immune evasion [81]. Several mechanisms are involved in promoting these events and initiate tumorigenesis. Molecular analysis revealed that specific regions of the virus are able to directly interact with the host transcription factor binding sites orchestrating regulatory regions used by tumor cells to control immune response [82]. Regarding their immunomodulatory nature, these events impair the activation of neutrophils, NKs, and TILs cells by dampening the expression of IFN- and IFN-related proteins in both innate and adaptive immunological responses, that include the activation of non-canonical signaling pathways such as the mitogen-activated protein kinase (MAPK) [83], phosphatidylinositol 3-kinases (PI3K) [84], nuclear factor kappa B (NF-κB) pathways [85], as well as the signal transducer and activator of transcription 3 (STAT3) [86], that prolongs the expression of a subset of interferon-stimulated immune regulatory genes.

In HPV+ HNSCC, a higher M1/M2 TAM ratio can be observed [87,88]. M1 macrophages are associated with a better prognosis and survival rate, whereas M2 phenotype is one of the key determinants of tumor progression and treatment failure [89]. This is partially explained because HPV can modulate MHC class I on the cell surface of antigen-presenting cells (APCs), including TAMs, which would impair the viral protein presentation to cytotoxic cells [22]. Furthermore, tissue resident dendritic cells (DCs) are essential for immune surveillance and act as qualified APCs to the effector cells [90,91]. Due to their plasticity and the presence of multiple receptors on their surface, DCs crosstalk with all cells in the immune system, and are critical for the initiation of anti-viral and antigen-specific immune responses [92]. However, to the best of our knowledge, DCs have not been considered a valid prognostic factor in HPV+ HNSCC.

In general, HPV+ HNSCC show significantly higher levels of TILs, especially CD8+ T cells [11]. Circulating T cells are constantly recruited to the TME in response to inflammatory signals after the recognition of antigen epitopes presented by APCs cells [93]. CD8+ T cells are detectable in 64–75% of HPV+ HNSCC samples [94,95,96]. These TILs produce pro-inflammatory cytokines (i.e., IFNγ and IL-17) with anti-tumoral activity that is related with the favorable prognosis in HPV+ HNSCC [97,98,99]. Studies have demonstrated that the quantity and quality of the immune infiltrate is a valid predictive tool that may improve the stratification of HNSCC patients [26]. Both HPV+ and HPV-negative HNSCC are infiltrated with Treg cells and NK cells overexpressing CD56^dim^ [100,101]. It was observed that different NK subsets are detected in HPV+ and HPV-negative tumors [102,103]. A common mechanism used by the virus to evade the host immune system is the reduction of MHC type I expression to escape a cytotoxic reaction. However, the specific role of NK cell-controlling HPV+ HNSCC is still under investigation [104]. This landscape provides a rationale for the investigation of agents targeting modulators of Tregs (e.g., CTLA-4, GITR, ICOS, IDO, and VEGFA) and NK cells (e.g., KIR, TIGIT, and 4-1BB) as adjuncts to anti–PD-1 in the treatment of advanced HNSCC [103].

## 4. The Impact of Therapeutic Schemes on the Immune Status of HNSCC

The main treatment for HNSCC includes surgery or radiation for the early-stage disease [105]. For recurrent/metastatic diseases, cytotoxic-based chemotherapy remains the standard therapeutic option and the median survival of HNSCC patients treated with palliative chemotherapy alone ranges from 6 to 10 months [106,107,108]. The combination of immunotherapeutic strategies represents a challenging approach, with a view to enhance anti-tumor immunity by targeting several aspects of the immune response [109]. The majority of HPV+ HNSCC patients have a favorable prognosis, and this raises the discussion about a less intensive treatment in order to decrease the side effects and improve patients’ quality of life. To date, several clinical trials have been proposed; however, few of them consider the HPV status for a personalized approach to target the TME dynamics (Table 1).

Furthermore, radiotherapy alone is known to induce substantial changes in the immune microenvironment in solid tumors [110]. Radiotherapy can control tumor growth by inducing cell death via direct DNA damage or generating reactive oxygen species (ROS) [110,111,112]. Although these mechanisms are important to kill cancer cells, radiation leads to the death of adjacent normal tissues causing severe adverse effects [110,113,114,115]. The radiotherapy influences the regulation of macrophage polarization [116], DCs phagocytosis [117], antigen intracellular processing and presentation to effector T cells [117,118,119], NK cell activation [120], as well as the cytokine and chemokine release [121]. The response of immune cells to radiation can determine the outcome of tumor therapy [122]. Recently, a large number of experimental and clinical trial studies have been conducted to manipulate the immune system, aiming to enhance the therapeutic efficiency of radiotherapy [122]. The clinical benefit can be observed in a substantial fraction of HNSCCs treated with immune checkpoint inhibitors, but the majority of tumors remain treatment-resistant [102]. Deciphering the basic mechanisms of upfront treatment resistance will require a detailed understanding of the immune infiltrative landscape of these tumors.

Standard treatment for locally advanced HNSCC consists mainly of chemoradiation using docetaxel, cisplatin, and/or fluorouracil in an attempt to eradicate potential microscopic residual cancer cells and ultimately improve loco-regional control and survival [123]. However, the intensification of therapy for patients who did not respond to the treatment is not able to overcome biologically aggressive HNSCC [123]. Checkpoint inhibitors such as anti-PD-1 and anti-PD-L1 antibodies were shown to significantly improve disease-free survival and overall survival after the failure of platinum-based chemotherapy [124]. With the introduction of immune checkpoint inhibitors to the clinic, a new set of toxicities, specifically, immune-related adverse events, have emerged. The side effects range from minimal to lethal and require a completely different management approach. Ipilimumab, in particular, is associated with grade 3–5 toxicity in 10–45% of HNSCC patients, depending on the dose, and whether it was given as a single agent or in combination with other immune therapies, chemotherapies, or molecularly targeted therapies [125]. Several clinical trials are in progress to evaluate the utility of checkpoint inhibitors in different treatment settings (Table 1). A cost-benefit and quality-of-life analysis may address the true contribution of the chemoradiotherapy associated with checkpoint inhibitors as an advantage strategy to treat patients with advanced HNSCC.

**Table 1 cancers-14-05406-t001:** Clinical trials for the treatment of HNSCC patients targeting the immune system.

Target	NCT Number	Status	Interventions	Phases	Enrolled Patients, n	Period (Start Date–Completion Date)	URL Access	Related Articles with Results	HPV Status	Immune Dynamics Evaluation	
										Intratumor Microenvironment	Peripheral Blood Cells
Inflammatory cell subsets	NCT00210470	Completed	IRX-2 (multiple cytokines) Cyclophosphamide Indomethacin Zinc Omeprazole	2	27	2005/07–2012/03	https://ClinicalTrials.gov/show/NCT00210470 (accessed on 28 September 2022)	[126,127,128,129]	n.d. **	Increased infiltration of TILs (CD3+, CD4+, CD8+ and CD20+ B cells) and CD68+ macrophages in tumor microenvironment. Peritumoral accumulation of CD4+ T cells. Predominance of intratumor CD8+ over CD4+ T cells. Higher CD20+ cells were associated with decreased tumor size. Increased survival rates associated with intratumor CD3+ and CD20+ cells.	Decreased levels of naïve T cells (CD3+CD45RA+CCR7+), central memory T cells (CD3+CD45RA−CCR7+CD27+), B lymphocytes (CD19+CD3−CD14−) and NKT cells (CD3+CD16+CD56+).
p-53- expressing tumor cells	NCT00496860	Completed	ALT-801 (humanized soluble T-cell receptor directed against the p53-derived antigen fused to IL-2)	1	26	2007/05–2009/10	https://ClinicalTrials.gov/show/NCT00496860 (accessed on 28 September 2022)	[130]	n.d.	n.d. **	Increased number of IFN-γ+ cells. Elevated serum IFN-γ levels.
Phosphodiesterase type-5	NCT00843635	Completed	Tadalafil (phosphodiesterase 5 (PDE5) inhibitor)	n/a *	35	2008/09–2015/04	https://ClinicalTrials.gov/show/NCT00843635 (accessed on 28 September 2022)	[131]	n.d.	n.d.	Decrease in m-MDSC and Treg cells numbers. Significant downregulation of MDSCs and nFoxp3:cFoxp3 ratio. Increased CD8+ cell activation.
Intratumor reactive T-cells and endothelial cells	NCT00953849	Completed	Celecoxib (cyclooxygenase 2 inhibitor) Calcitriol (Vitamin D)	1|2	21	2009/11–2015/12	https://ClinicalTrials.gov/show/NCT00953849 (accessed on 28 September 2022)	none	n.d.	Intra-tumor increased IL-2, IFN-γ, and GM-CSF and decreased IL-6 staining.	n.d.
Tumor cells	NCT01302834	Unknown	cetuximab (anti-EGFR mAb) cisplatin (apoptosis-inducer via DNA crosslinking) MRT	3	987	2011/02–	https://ClinicalTrials.gov/show/NCT01302834 (accessed on 28 September 2022)	[132]	Yes	n.d.	n.d.
Interleukin-6	NCT01403064	Terminated	ALD518 (humanized anti-IL-6 antibody)	2	76	2011/07–2014/03	https://ClinicalTrials.gov/show/NCT01403064 (accessed on 28 September 2022)	none	n.d.	n.d.	n.d.
Anti-tumor cellular immunity	NCT01468896	Active, not recruiting	Cetuximab (anti-EGFR mAb) Edodekin alfa (recombinant IL-12)	1|2	23	2011/11–	https://ClinicalTrials.gov/show/NCT01468896 (accessed on 28 September 2022)	none	n.d.	n.d.	n.d.
HPV-infected cells and tumor cells	NCT01585428	Completed	Fludarabine (inhibitor of DNA synthesis) Cyclophosphamide (inhibitor of protein synthesis) Young TIL (Tumor Infiltrating Lymphocytes) Aldesleukin (recombinant IL-2)	2	29	2012/04–2016/08	https://ClinicalTrials.gov/show/NCT01585428 (accessed on 28 September 2022)	[133]	Yes	n.d.	n.d.
Phosphodiesterase type-5	NCT01697800	Completed	Tadalafil (phosphodiesterase 5 (PDE5) inhibitor)	2	40	2012/09–2014/07	https://ClinicalTrials.gov/show/NCT01697800 (accessed on 28 September 2022)	none	n.d.	n.d.	n.d.
Inflammation and pain	NCT01883908	Terminated	Acupuncture Usual medical care for pain relief	n/a	4	2012/12–2015/02	https://ClinicalTrials.gov/show/NCT01883908 (accessed on 28 September 2022)	none	n.d.	n.d.	n.d.
Innate and adaptive immunity crosstalk	NCT01984892	Terminated	Poly-ICLC (TLR3-ligand)	2	8	2013/11–2014/08	https://ClinicalTrials.gov/show/NCT01984892 (accessed on 28 September 2022)	[134]	n.d.	n.d.	n.d.
HPV-specific T cell repertoire	NCT02002182	Active, not recruiting	ADXS11-001/ADXS-HPV (immunobiological product from Listeria monocytogenes)	2	15	2013/12–2023/08	https://ClinicalTrials.gov/show/NCT02002182 (accessed on 28 September 2022)	none	Yes	n.d.	No difference between treatment and control groups on HPV-specific T cell response rate.
HPV-specific T and B cell repertoires	NCT02163057	Completed	INO-3112 (plasmids encoding HPV oncoproteins delivered by electroporation system)	1|2	22	2014/08–2017/01	https://ClinicalTrials.gov/show/NCT02163057 (accessed on 28 September 2022)	none	Yes	Suggestive modulation of CD8+, perforin+ and FoxP3+ TILs.	n.d.
HPV-infected cells and tumor cells	NCT02280811	Completed	Fludarabine (inhibitor of DNA synthesis) Cyclophosphamide (inhibitor of protein synthesis) E6 TCR (T cells genetically engineered with a TCR targeting HPV-16 E6 oncoprotein) Aldesleukin (recombinant IL-2)	1|2	12	2014/10–2016/06	https://ClinicalTrials.gov/show/NCT02280811 (accessed on 28 September 2022)	none	Yes	n.d.	Inconclusive results.
Anti-tumor cellular immunity	NCT02315066	Completed	PF-04518600 (OX40 agonist) PF-05082566 (4-1BB agonist)	1	174	2015/04–2020/11	https://ClinicalTrials.gov/show/NCT02315066 (accessed on 28 September 2022)	[135]	n.d.	Upregulation of gene sets associated with anti-tumor immune response, mainly IFN-γ-related pathways.	Increased CD4+ and CD8+ T-cell clonal expansion.
Anti-tumor cellular immunity	NCT02521870	Terminated	SD-101 (synthetic CpG oligonucleotide acting as TLR9 ligand) Pembrolizumab (programmed death receptor-1 (PD-1)-blocking antibody)	1|2	241	2015/09–2020/04	https://ClinicalTrials.gov/show/NCT02521870 (accessed on 28 September 2022)	[136]	n.d.	n.d.	n.d.
Lymph system	NCT03332160	Completed	Flexitouch (pneumatic compression device)	n/a	49	2018/01–2019/07	https://ClinicalTrials.gov/show/NCT03332160 (accessed on 28 September 2022)	none	n.d.	n.d.	A slight decrease in IL-6 levels.
Intratumor reactive T-cells	NCT03463161	Terminated	Pembrolizumab (programmed death receptor-1 (PD-1)-blocking antibody) Epacadostat (selective inhibitor of indoleamine 2,3-dioxygenase 1 (IDO1)	2	2	2018/03–2018/12	https://ClinicalTrials.gov/show/NCT03463161 (accessed on 28 September 2022)	none	Yes	n.d.	n.d.
Intratumor reactive T-cells	NCT03938337	Terminated	Pembrolizumab (programmed death receptor-1 (PD-1)-blocking antibody) Abemaciclib (inhibitor of cyclin-dependent kinases (CDK))	2	1	2019/10–2020/04	https://ClinicalTrials.gov/show/NCT03938337 (accessed on 28 September 2022)	none	n.d.	n.d.	n.d.
HPV-infected cells and tumor cells	NCT04015336	Terminated	E7 TCR (T cells genetically engineered with a TCR targeting HPV-16 E7 oncoprotein)	2	1	2020/06–2020/07	https://ClinicalTrials.gov/show/NCT04015336 (accessed on 28 September 2022)	none	Yes	n.d.	n.d.
Intratumor reactive T-cells and NK cells	NCT04099277	Terminated	LY3435151 (anti-CD226) Pembrolizumab (programmed death receptor-1 (PD-1)-blocking antibody)	1	2	2019/10–2020/03	https://ClinicalTrials.gov/show/NCT04099277 (accessed on 28 September 2022)	none	n.d.	n.d.	n.d.
Intratumor reactive T-cells	NCT01848834	Completed	Pembrolizumab (programmed death receptor-1 (PD-1)-blocking antibody)	1	297	2013/05–2020/06	https://ClinicalTrials.gov/show/NCT01848834 (accessed on 28 September 2022)	[137,138,139,140,141]	Yes	n.d.	n.d.
Intratumor reactive T-cells	NCT03083873	Completed	LN-145 (autologous TIL-mediated adoptive cell transfer therapy) recombinant IL-2 non-myeloablative (NMA) lymphodepletion	2	112	2017/01–2022/03	https://clinicaltrials.gov/ct2/show/NCT03083873 (accessed on 28 September 2022)	none	Yes	n.d.	n.d.

Data source: adapted from ClinicalTrials.gov. * n/a = not applicable. ** n.d. = not described.

## 5. Treatment Strategy and Vaccine for Patients with HPV+ HNSCC

The determination of the HPV status may guide clinicians in their prognostic assessment and treatment decision-making in the HNSCC population [142]. HPV+ HNSCC has better outcomes and is more sensitive to radiotherapy and chemotherapy compared with HPV-negative HNSCC, which may be due to the effective immune responses to viral and abundant numbers of infiltrating immune cells. The fact that HPV+ HNSCC has a good prognosis provides the rationale for several clinical trials with de-intensified treatment or alternative therapeutic approaches. De-intensification strategies involve less invasive surgery, such as transoral robotic surgery (TORS), which utilizes miniaturized instruments to perform the resection of selected cancer areas, as well as a reduction in the dose of chemotherapy and/or radiotherapy [143].

The adoption of anti-viral strategies to combat HPV infections, including anti-HPV vaccines, might also modulate the TME and influence the tumor response. HPV vaccines have a clear role in preventing cervical cancer and conditions related to HPV infection [66,144]. The prophylactic vaccines recommended by the FDA are bivalent (HPV16 and HPV18), quadrivalent (HPV6, HPV11, HPV16, and HPV18), or nine-valent (HPV6, HPV11, HPV16, HPV18, HPV31, HPV33, HPV45, HPV52, and HPV58) vaccines. Recently, the FDA and Health Canada approved an expanded indication for the HPV nine-valent vaccine for the prevention of HNSCC [144]. There is hope that preventive HPV vaccinations can also reduce the occurrence of HPV+ HNSCC. Several long-term trials are underway to evaluate their effectiveness and to understand how these vaccines modulate the anti-viral immunity. The goal of cancer vaccination is to obtain anti-cancer effects by activating or increasing an effective CD4+/CD8+ antigen-specific T cell response [145].

HPV vaccines commercially available to date (Gardasil, Gardasil 9, and Cervarix) are able to induce immune response by blocking the viral fusion and entry into the host cell [146]. Specifically, all of these vaccines are designed with VLPs from the HPV structural protein L1 [55], which stimulates naive B cells and increases antibody production [146,147]. These vaccines are now in clinical trials for HPV-driven cancers including HNSCC. However, the prophylactic HPV against L1 proteins appears to be ineffective in the treatment of HPV-induced cancers [148]. New HPV vaccines have been designed to target the oncoproteins E6 and E7 because they are constantly necessary and exclusively produced in cancer cells [146,149]. The challenge for therapeutic oncological vaccines is to stimulate an immune T cell response to endogenous antigens that is sufficiently potent to induce cytotoxic activity and broad enough to take tumors and TME heterogeneity into account.

## 6. Conclusions

In this review we discussed how the presence of HPV interferes with the local and systemic immune response that may lead to a complete response or resistance to the treatment and consequently impact on the prognosis and survival of patients with HNSCC. Understanding the immunological dynamic associated with tumor cell behavior is the key to developing novel immunotherapeutic targets and strategies to treat HNSCC.

## Figures and Tables

**Figure 1 cancers-14-05406-f001:**
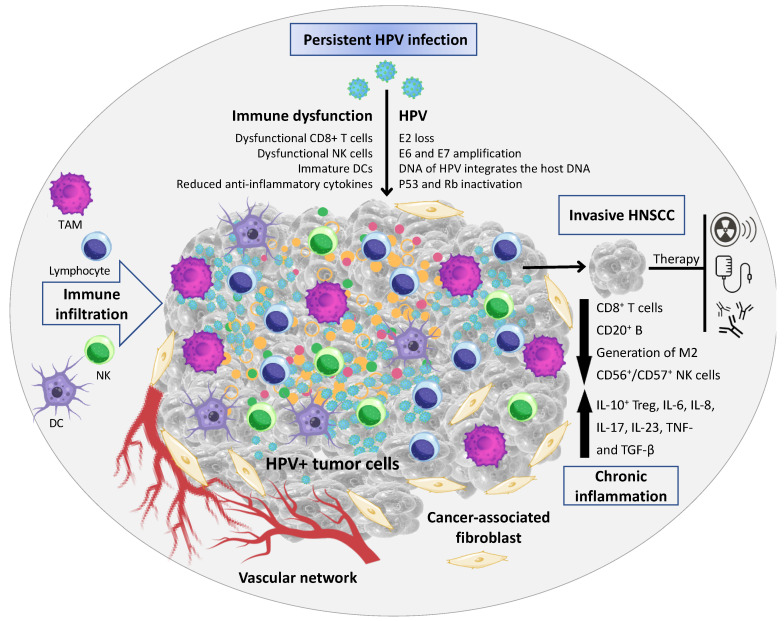
Tumor microenvironment (TME) associated with HPV+ HNSCC. The TME comprises malignant epithelial cells, and a heterogeneous cell population integrated in a complex extracellular matrix (ECM) 16. The main cellular components of the TME are tumor-infiltrating lymphocytes (a.k.a.: TILs; or B and T lymphocytes), tumor-associated macrophages (TAMs), natural killer cells (NKs), tumor-associated neutrophils (TANs), dendritic cells (DCs), and cancer-associated fibroblasts (CAFs). In HPV+ HNSCC the virus has a key role in the immune dysfunction by the recruitment and activation of cytokines- and chemokines-regulating cells associated with tumor growth and dissemination (Image created using Canva Pro Software at https://www.canva.com/pro/, accessed on 28 September 2022).

**Figure 2 cancers-14-05406-f002:**
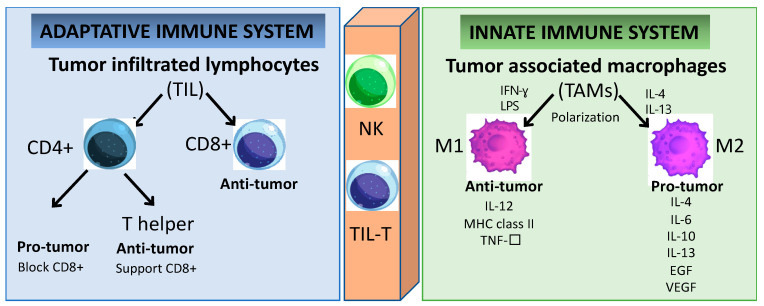
Immune cells play a key role in tumor cell growth and dissemination. Cells of the innate immunity branch provide a rapid response to non-self-antigens. In contrast, cells of the adaptive immunity branch provide a slower but specific response. Several cell subsets, including TIL and NK cells, connect both branches of immunity because they express receptors similar to those in conventional B and T cells. Even though the specificity of these receptors is limited, the response to specific non-self-antigens is prompt.

**Figure 3 cancers-14-05406-f003:**
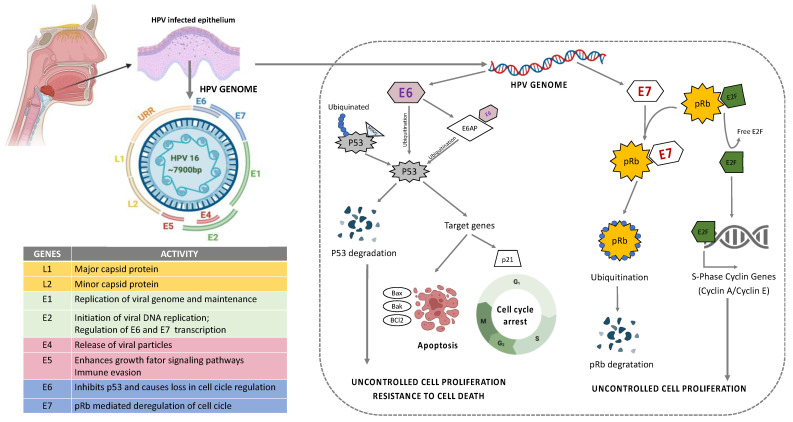
Structure and organization of HPV16 genome. E6 mediated p53 manipulation and E7 mediated inhibition of pRb protein leading to sustained cell proliferation and resistance to apoptotic barrier.

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
