# Peer review of "Implications and Emerging Therapeutic Avenues of Inflammatory Response in HPV+ Head and Neck Squamous Cell Carcinoma"

_cancers, 2022, doi:10.3390/cancers14215406_

Round 1

Reviewer 1 Report

This is a reasonably comprehensive review of the inflammatory/immune response to HPV+ OPSCC.  There are some misunderstandings and inaccuracies that should be corrected.  

There are more than 200 (not100) HPV types that have been isolated

L1 and L2 are the major and minor capsid or coat proteins of the virus.   The virus like particle is NOT the virus (page 5 para 2)

HPV entry into cells is still controversial why describe CD16 mediated entry into NK cells reference other putative receptors but not describe this in the text

In contrast to the statements made by the authors (page 6 para1) integration of viral DNA into the host genome is not a prerequisite for cancerous transformation, many HPV+ OPSCC have episomal HPV.  Furthermore virus does not integrate into the host genome; viral DNA sequences integrate into the host genome 

The epithelium in the tonsillar crypts is quite different to the genital tract squamous epithelia, this should be discussed in a review on HNSCC and the response to HPV

Author Response

REVIEWERS' COMMENTS

We thank the reviewer for their valuable comments, which we have addressed and believe have strengthened the quality of our manuscript. Please see below our point-by-point responses and the corrections we made in the revised manuscript (tracked in yellow for your convenience).

REVIEWER #1:

There are more than 200 (not100) HPV types that have been isolated.

Response: This information was revised in the main text (pages 1 and 2).

L1 and L2 are the major and minor capsid or coat proteins of the virus. The virus like particle is NOT the virus (page 5 para 2).

Response: We apologize for this inadvertent mistake. The sentence was rewritten (first paragraph - page 5).

HPV entry into cells is still controversial why describe CD16 mediated entry into NK cells reference other putative receptors but not describe this in the text.

Response: As suggested, we discussed in details the controversy of HPV entry into cells (paragraph 1 – page5).

In contrast to the statements made by the authors (page 6 para1) integration of viral DNA into the host genome is not a prerequisite for cancerous transformation, many HPV+ OPSCC have episomal HPV. Furthermore, virus does not integrate into the host genome; viral DNA sequences integrate into the host genome

Response: We greatly appreciate this comment. We have rewritten this sentence to avoid misunderstanding about HPV intergration (first paragraph - page 6). We also added the information and references regarding episomal HPV (first paragraph – page 6).

The epithelium in the tonsillar crypts is quite different to the genital tract squamous epithelia, this should be discussed in a review on HNSCC and the response to HPV

Response: We agree with the reviewer and the information was added (first paragraph – page 2).

Reviewer 2 Report

It is an interesting work to help us understand the complexity of the inflammatory networks and the mechanisms of immune evasion in HPV+ HNSCC to find more useful therapeutic methods. 

I only have a small suggestion. In this paper, the authors have mentioned that the intratumor immune dynamic in HPV+ HNSCC is different compared to HPV-negative HNSCC. Understanding the more detail of intratumor immune dynamic in HPV+ HNSCC is very important to find more useful therapeutic methods. So I think that if more detail on the tumor microenvironment of Clinical trials is provided and added in Table 1 will be better. 

Author Response

REVIEWERS' COMMENTS

We thank the reviewer for their valuable comments, which we have addressed and believe have strengthened the quality of our manuscript. Please see below our point-by-point responses and the corrections we made in the revised manuscript (tracked in yellow for your convenience).

REVIEWER #2:

It is an interesting work to help us understand the complexity of the inflammatory networks and the mechanisms of immune evasion in HPV+ HNSCC to find more useful therapeutic methods.

I only have a small suggestion. In this paper, the authors have mentioned that the intratumor immune dynamic in HPV+ HNSCC is different compared to HPV-negative HNSCC. Understanding the more detail of intratumor immune dynamic in HPV+ HNSCC is very important to find more useful therapeutic methods. So I think that if more detail on the tumor microenvironment of Clinical trials is provided and added in Table 1 will be better.

Response: Thank you for this suggestion. We undertook a comprehensive review of clinical trials in relation to tumor microenvironment and a summary information is included as a new column in Table 1 of the manuscript. A new table (below) is also added in the revised version of the manuscript.
